# Efficacy of Lymph Node Location-Number Hybrid Staging System on the Prognosis of Gastric Cancer Patients

**DOI:** 10.3390/cancers15092659

**Published:** 2023-05-08

**Authors:** Junpeng Wu, Hao Wang, Xin Yin, Xibo Wang, Yufei Wang, Zhanfei Lu, Jiaqi Zhang, Yao Zhang, Yingwei Xue

**Affiliations:** Department of Gastroenterological Surgery, Harbin Medical University Cancer Hospital, Harbin Medical University, Harbin 150081, China

**Keywords:** gastric cancer, N staging, lymph nodes metastasis, prognosis

## Abstract

**Simple Summary:**

Lymph node staging is very important for the prognosis of patients with gastric cancer. Currently, the internationally accepted lymph node staging method is the 8th AJCC staging, which divides lymph nodes into different stages according to the number of positive lymph nodes. This staging method is simple and convenient, but ignores the laterality of lymph nodes. In this paper, we for the first time combined the location and number information of positive lymph nodes to create a novel lymph node staging system for gastric cancer. After training cohort and validation cohort tests, this staging is more accurate in predicting the prognosis of patients than the 8th AJCC staging.

**Abstract:**

Background: Lymph node metastasis location and number significantly affects the prognosis of patients with gastric cancer (GC). This study was designed to examine a new lymph node hybrid staging (hN) system to increase the predictive ability for patients with GC. Methods: This study analyzed the gastrointestinal treatment of GC at the Harbin Medical University Cancer Hospital from January 2011 to December 2016, and selected 2598 patients from 2011 to 2015 as the training cohort (hN) and 756 patients from 2016 as the validation cohort (2016-hN). The study utilized the receiver operating characteristic curve (ROC), c-index, and decision curve analysis (DCA) to compare the prognostic performance of the hN with the 8th edition of AJCC pathological lymph node (pN) staging for GC patients. Results: The ROC verification of the training cohort and validation cohort based on each hN staging and pN staging showed that for each N staging, the hN staging had a training cohort with an AUC of 0.752 (0.733, 0.772) and a validation cohort with an AUC of 0.812 (0.780, 0.845). In the pN staging, the training cohort had an AUC of 0.728 (0.708, 0.749), and the validation cohort had an AUC of 0.784 (0.754, 0.824). c-Index and DCA also showed that hN staging had a higher prognostic ability than pN staging, which was confirmed in the training cohort and the verification cohort, respectively. Conclusion: Lymph node location-number hybrid staging can significantly improve the prognosis of patients with GC.

## 1. Introduction

According to the World Health Organization (WHO), gastric cancer (GC) was the fifth common global tumor incidence in 2020, and its mortality rate was ranked fourth, as it is responsible for approximately 770,000 deaths per year [1]. Due to the wide distribution of gastric lymph nodes and complex drainage [2], GC becomes a physical tumor with a high tendency to metastasize lymph nodes [3]. In many countries the proportion of lymph node metastasis is between 54–64% [4,5,6,7]. Even if R0 resection combined with D2/D2+ cleaning is performed on patients with lymph node metastasis, their survival period is even worse than those with no lymph node metastasis [7,8,9]. Recently, N staging has been updated to the 8th edition AJCC staging. Compared with the 7th edition, the 8th edition AJCC has a slight adjustment of the N staging; in pathological classification, N3 was further subdivided into N3a (7–15 positive lymph nodes) and N3b (≥16 positive lymph nodes) [10]. In this way, a more accurate prognosis assessment and personalized follow-up treatment for GC patients can be obtained [2,10].

Although the number of positive lymph nodes can predict the prognosis of patients, Daniele et al. [11] found that patients with positive lymph nodes (8p, 12b/p, 13, etc.) in the rear group had a worse prognosis than those with negative lymph nodes. It was also shown that the location of lymph node metastasis significantly impacts the prognosis of patients. The limitation of quantitative staging is that it does not provide anatomical information about lymph nodes and may also lead to inconsistent preoperative and postoperative N staging [12,13]. In order to improve prognostic efficacy, Choi et al. [12] proposed a hybrid lymph node staging. The stomach lymph nodes were divided into three parts (lesser curvature LN groups, greater curvature LN groups, and extra-perigastric LN groups), only focusing on the location of positive lymph nodes, and not the number of positive lymph nodes. This site-specific lymph node staging was demonstrated to have the same predictive performance as the 7th edition AJCC pN staging at low infiltration depth. The simplicity of this staging makes it an alternative to the 7th edition AJCC and helps in improving the consistency of preoperative and postoperative staging in patients with GC [14]. Subsequently, it was further improved in two single-center studies in Taiwan [15] and Italy [16]. Choi et al. [12] determined that hN staging could be used as an alternative to pN staging in the 7th edition. However, the above solutions still ignore the effect of the number of metastatic lymph nodes on the prognosis of patients [17,18,19]. The comparison content at the 7th edition AJCC, and the effect of the 8th edition is still unknown.

This study proposed a new hybrid lymph node staging through retrospective analysis and combined it with the advantages of location and number staging. In addition, the prediction ability and accuracy of the two systems were compared to propose a supplementary standard that can be used as the existing pN staging.

## 2. Patients and Methods

### 2.1. Patients

The diagnosis of GC was based on the tissue samples obtained using preoperative gastroscopy and further confirmed via histopathological examination by a professional pathologist. In addition, the patient underwent routine preoperative examinations, including abdominal CT, chest CT, echocardiography, ultrasound of bilateral supraclavicular lymph nodes, electrocardiogram, routine hematology, and tumor marker examination. Every patient had more than 16 lymph node cleanings. Following lymph node removal, each patient was photographed and sent to the pathology lab separately. At least two pathology experts reviewed all pathological results. All staging was conducted in accordance with the 8th edition AJCC.

A total of 4712 surgical patient in the Department of Gastrointestinal Surgery of Harbin Medical University Cancer Hospital from 1 January 2011, to 31 December 2016, were selected for this study. Patients’ clinicopathological data were saved in the GC information management system v1.2 of Harbin Medical University Cancer Hospital (Copyright 2013SR087424, http://www.sgihmu.com, accessed on 10 August 2022). Gender, age, tumor diameter, tumor location, pTNM stage, venous invasion, nerve invasion, and postoperative chemotherapy were included. Patients with negative surgical margins (R0) and standard D2/D2+ surgery were enrolled. The inclusion criteria for our study were as follows: ① patients who underwent radical gastrectomy in the Affiliated Cancer Hospital of Harbin Medical University between 1 January 2011, to 31 December 2016; and ② had regular follow-up for at least 5 years. Exclusion criteria included: ① history of the other malignancies; ② patients with preoperative neoadjuvant chemotherapy; ③ the postoperative pathological report was non-tumor; ④ patients with unclear lymph node metastasis or invasion depth reported by postoperative pathology; and ⑤ patients with a history of gastric surgery. A total of 3354 patients were included after the exclusion. Finally, 2598 patients were chosen as the training cohort (hN) from January 2011 to December 2015, and 756 patients were chosen as the validation cohort (2016-hN) from February 2016 to November 2016.

### 2.2. Follow Up

All patients were followed up after discharge by telephone, e-mail, or examination at the Affiliated Cancer Hospital of Harbin Medical University outpatient complex through the analysis of hematology, tumor markers, gastroscopy, abdominal ultrasonography, abdominal CT, and PET-CT for some patients according to their condition. Stage I patients were followed up every 12 months, stage II patients every 6 months, and stage III patients every 3–6 months. In addition, chest or abdominal CT was performed for suspected tumor recurrence or elevated tumor marker levels above pathological levels, and a bone scan was performed for suspected bone metastases. The diagnosis of recurrence is confirmed based on imaging or re-surgical pathology.

### 2.3. Statistics Analysis

Statistical analyses were performed using IBM, SPSS, and R. Chi-square or Fisher’s exact test was used for categorical variables. A Chi-square test was performed to analyze the relationship between recurrence and the clinicopathological characteristics of the patients. Factors with significant differences (*p* < 0.05) that appeared when using univariate analysis were then entered into multivariate analysis. The optimal cut-off value of the number of positive lymph nodes was calculated using X-tile. The receiver operating characteristic curve (ROC) was used to calculate the accuracy of the recurrence scores in predicting recurrence and the area under the curve (AUC). Harrell Consistency Index (C-index) analysis and DCA test were performed using the survival package in R.

### 2.4. Lymph Nodes Location-Number Hybrid Staging System Criteria

According to the classification of gastric lymph nodes by Choi et al. [12] (Figure 1), gastric lymph nodes are divided into two groups according to their anatomical positions: Perigastric lymph nodes and extra-perigastric lymph nodes (EP). Of these, perigastric lymph nodes were divided into lesser curvature LN groups (LC, including 1/3/5 groups lymph nodes) and greater curvature LN groups (GC, including 2/4/6 groups lymph nodes). EP included all lymph nodes except those mentioned above and para-aortic lymph nodes. Based on Choi et al. [12], we grouped the positive lymph nodes of the EP group according to the number, calculated the optimal cut-off value of the number of positive lymph nodes through X-tile software to ensure a good layering effect in each staging period and determined the best combination method. The final hN stage was as follows:

hN0: No positive lymph nodes.hN1: LC/GC had positive lymph nodes irrespective of the number, or EP had positive lymph nodes, and the number of positive lymph nodes in the EP group was ≤4.hN2: Two groups of LC/GC/EP had positive lymph nodes, regardless of the number. EP had positive lymph nodes, and the number of lymph nodes was >4.hN3a: LC + GC + EP all had positive lymph nodes, and the total number of positive lymph nodes was ≤14.hN3b: LC + GC + EP all had positive lymph nodes, and the total number of positive lymph nodes was >14.

## 3. Results

### 3.1. Patient Baseline Characteristics

A total of 3354 patients were included in this study. Table 1 shows the patient characteristics. Here, 1899 patients were ≤60 years old, and 1455 patients were >60 years old. There were 2444 males and 910 females. According to the hN stage, the number of patients with hN0, hN1, hN2, hN3a, and hN3b was 1671 (49.82%), 587 (17.50%), 588 (17.53%), 280 (8.35%), and 228 (6.80%), respectively. According to the hTNM staging, 1154 (34.41%) patients were in stage I, 1029 (30.68%) in stage II, and 1171 (34.91%) in stage III. Compared with the validation cohort, the training cohort had statistically significant differences in the hN stage, pT stage, pathological type, degree of differentiation, HER-2, BMI, CEA, and tumor size. However, there was no significant difference in age, gender, tumor location, hTNM stage, CA-199, lymph node metastatic ratio (LNRs), number of metastatic lymph nodes (MLNs), and the total number of removed lymph nodes (RLNs).

From 2011 to 2015, 2598 patients were included in the training cohort, including hN0 1291 (49.69%), hN1 483 (18.59%), hN2 443 (17.05%), hN3a 223 (8.58%), and hN3b 158 (6.08%). The average age was 57.97 ± 10.05. There were 1909 males (73.48%) and 689 females (26.52%) (*p* = 0.567). Most tumors were located in the lower part (1890, 72.75%), 403 (15.51%) followed by the middle part, 269 (10.35%), then the upper part, and 36 (1.39%) in the whole stomach. From hN0–hN3b, patients with tumors in the lower part decreased gradually, while patients with tumors in the middle, upper part, and whole stomach increased gradually. The proportion of hN0–hN3b decreased gradually from T1 to T2 and increased gradually from T3 to T4. BMI, CEA, CA-199, RLNs, MLNs, and RLNs showed significant differences under different hN stages (Table 2).

A total of 756 patients in 2016 were included in the validation cohort. According to the hN staging, the proportion of 2016-hN0, 2016-hN1, 2016-hN2, 2016-hN3a, and 2016-hN3b was 380 (50.26%), 104 (13.76%), 145 (19.18%), 57 (7.54%), and 70 (9.26%), respectively. Furthermore, 405 (53.57%) patients were less than 60 years, and 351 (46.43%) patients were more than 60 years old (*p* = 0.859). With the progress of the hN stage, the proportion of tumors located in the lower part of the stomach gradually decreased, while the proportion of tumors located in the upper part and the whole stomach gradually increased. BMI, CEA, CA-199, RLNs, MLNs, and RLNs showed significant differences under different hN stages (Table 3).

### 3.2. Evaluation of the Predictive Ability of hN Staging

According to the ROC test, the AUC of hN and pN for the training cohort were 0.752 (95%CI: 0.733–0.772) and 0.728 (95%CI: 0.708–0.749), respectively. Furthermore, the AUC of 2016-hN and 2016-pN for the validation cohort were 0.812 (95%CI: 0.780–0.845) and 0.784 (95%CI: 0.754–0.824), respectively (Figure 2a,b).

For hN and pN, C-index was calculated under the conditions of differentiation degree (Table 4a), T stage (Table 4b), and TNM stage (Table 4c). hN C-index was greater than pN at all differentiation degrees. For example, in G1, G2, and G3, hN C-index was 0.718, 0.683, 0.750, pN C-index was 0.712, 0.671, 0.737, respectively. In the T1-T4 stage, the C-index of hN in T1, T2, T3, and T4 was 0.550, 0.703, 0.742, and 0.674, respectively. The C-index of pN was 0.531, 0.648, 0.689, and 0.660, respectively. For TNM staging, the C-index of hN was the same as that of pN in stage I (0.520). However, for stages II and III, the C-index of hN was significantly different from that of pN. The C-index of hN in stages II and III was 0.623 and 0.621, respectively. The C-index of pN in stages II and III was 0.611 and 0.606, respectively.

DCA results showed that in the validation cohort, hN staging and hTNM staging were better than pN and pTNM in the 3rd and 5th years (Figure 3a–d), with a higher area under the curve. In addition, the results of the training cohort were well verified in the validation cohort (Figure 3e–h).

The Kaplan–Meier curves of the 5-year overall survival (OS) rate of each subgroup under the condition of hN in the training cohort and the validation cohort are shown in Figure 4a,b, with *p* < 0.01 among each stratum. The hN stage had significant statistical differences under each N stratum, indicating that the hN stage had a good stratification (Figure 4)

### 3.3. Univariate and Multivariate Analysis of Prognostic Factors in Patients with hN Staging

In order to determine the independent risk factors affecting the prognosis of patients under the hN stage, Cox risk regression model analysis was performed. Univariate analysis showed that BMI, CEA, CA-199, MLNs, RLNs, LNRs, age, tumor location, hN stage, pT stage, hTNM stage, pathological tumor type, degree of differentiation, and tumor size were statistically significant. Multivariate analysis showed that BMI, CEA, CA-199, RLNs, age, tumor location, hN stage, pT stage, hTNM stage, and pathological type were independent risk factors related to the prognosis of patients (Table 5). Based on multivariate regression analysis, a prognostic Nomogram was established according to independent prognostic factors (Figure 5) to show the relationship between each predictor variable in the prediction model.

## 4. Discussion

Lymph node metastasis is one of the most critical factors affecting the survival of patients with GC, as it can lead to the recurrence of GC [17,18,19]. Therefore, selecting a staging system with good predictive power is essential. In this study, we also considered the location and number of lymph node metastases and constructed hN staging and found that hN staging had better prognostic prediction performance and value than pN staging. In addition, hN staging also allows patients to survive with good discrimination.

Mine et al. [21] found that different tumor locations significantly affect the prognosis and lymph node metastasis of patients with esophageal cancer. In addition, different lymph node metastasis sites also significantly impact the prognosis of patients, which is called tumor laterality. The complexity of the arteries supplying blood to the stomach and the surrounding areas results in severe challenges in adequately identifying the anatomy of the lymphatic system. Therefore, there is no method for staging based on the anatomical location of lymph nodes [2]. We speculate that laterality may have a potential effect on GC, though the effect of tumor location and lymph node metastasis laterality on N staging has not been reported in GC [22]. In addition, GC is a highly heterogeneous malignant tumor, and the same N stage may have different prognosis for GC patients. Therefore, considering the hN staging may further differentiate the prognosis of patients with GC.

In order to solve the above problems, Choi et al. [12] created a new regional lymph node staging system based on the location of positive lymph nodes, which made up for the lack of lymph node laterality in the current staging. They showed that the number of lymph node metastasis was 2.2 ± 1.8, 2.3 ± 2.0, and 2.1 ± 1.8 for simple lesser curvature, greater curvature, and peripheral lymph node metastasis, respectively. In this study, the number of simple lesser curvature, greater curvature, and peripheral lymph node metastasis was 2.4 ± 2.3, 2.1 ± 1.7, and 2.1 ± 2.2, respectively, at *p* = 0.036. It was found that there was a significant difference in the number of metastatic lymph nodes in different sites. The number of metastatic lymph nodes in the greater curvature was significantly higher than in the lesser curvature and peripheral lymph nodes. Previous studies have shown that the metastatic ability of lymph nodes is related to the distance of the lymph nodes from the tumor. According to the distance from the tumor location to the lymph nodes, the tumor in the greater curvature is more likely to metastasize to the 6 group lymph nodes outside the greater curvature [23], which is more likely to metastasize to the 14v and 16 groups of lymph nodes [24,25]. Moreover, Jung et al. [22] found that the prognosis ability of patients with tumors located at the greater curvature was worse than that at other parts, and it was speculated that the reason was also due to the stronger migration ability of lymph nodes located at the greater curvature. Therefore, staging based solely on positive lymph node location may lead to a decline in the ability to judge the prognosis of the disease. Because the number of positive lymph nodes at different sites leads to different lymph node migration abilities, which significantly impacts the prognosis of patients, we added the number of lymph nodes to control this difference and obtain better predictive ability.

The latest 8th AJCC staging has improved the predictive performance of pTNM staging for the prognosis of different patients [26]. However, there are still differences in conclusions in different countries [27]. The main reason is that the current staging ignores the value of positive lymph node location on the prognosis of patients [11]. In addition, the regional differences in surgical methods also impact the results of different countries. Asian countries, mainly Japan and Korea, recommend D2/D2+ lymph node dissection for patients with advanced GC [28,29,30,31]. Whereas in Europe, D1 dissection is preferred for patients with curable GC [6,32]. In recent years, scholars worldwide have proposed various lymph node staging methods, such as metastatic lymph node ratio [33,34,35] and negative lymph node/T stage log [36], to adjust N staging. However, these two methods have limitations. The primary disadvantage of using metastatic lymph node ratio [33,34,35] as the N staging standard is that its staging range is unclear. There are significant differences in the lymph node metastasis rate staging in different regions, and the definition of different staging critical points is controversial [12], which is why it cannot be used as a universal staging system. The N staging of the logarithm of negative lymph nodes/T stage [36] has not been supported by the literature in other countries. The above two staging methods rely on the number of positive lymph nodes and cannot account for the impact of lymph node location on prognosis.

It was found in this study that the C-index of hN was higher than that of pN except for TNM I patients. In stage I patients, the C-index of hN and pN was the same, which was 0.520, and the C-index was between 0.5–0.6, showing poor predictive ability. The possible reason is that according to the 8th edition AJCC staging criteria [10], pTNM I includes N0-T1, N0-T2, and N1-T1, of which 743 patients were N0-T1, 334 patients were N0-T2, 77 patients were hN1-T1, and 65 patients were pN1-T1. N1-T1 accounted for 5.69–6.67% of the total TNM I, and due to the low number of lymph node metastases in stage I patients, staging based on the location and number of lymph node metastases may not accurately predict the prognosis. However, for patients with N2 and later, the accuracy of hN staging is improved due to the increase in the number of lymph node metastases. Therefore, the C-index of the hN and pN stages was the same at the TNM II stage. The new hN staging showed better prediction accuracy than pN staging in T1–T4, G1–G3, and TNM II–III stages and the prediction precision was higher (Table 4a–c).

In this study, we found that with the gradual progress of the hN stage, the proportion of tumors located in the lower part gradually decreased, while the proportion of tumors located in the gastric body, cardio, and whole stomach gradually increased (Table 1, Table 2 and Table 3). Therefore, Li et al. [37] proposed that the distribution of GC tumors would change over time. According to reports, among Korean GC patients, the proportion of distal lesions gradually decreases during tumor progression [38], which may be due to the anatomical differences in the stomach. Compared with the gastric antrum, the thickness of the gastric wall in the upper part of the stomach is thinner, and the thickness of the submucosa is also thinner. Therefore, GC located in the upper part of the stomach is more likely to cause tumor lymph node metastasis [39]. In addition, the blood supply is more abundant than other sites, so distant lymph node metastasis is more likely to occur. Because the hN staging includes the location of lymph node metastasis in the N staging diagnostic criteria, and the location of the tumor is related to the location of the lymph node metastasis, therefore, the hN staging is related to the tumor’s location.

Verlato et al. [40] found that the metastatic lymph nodes are the most important prognostic factors in patients with GC. However, their prognostic significance often diminishes over time. The mortality peak of GC patients is related to the N stage, and the higher the stage, the more the mortality peak. The peak mortality caused by the N stage generally occurred within 24 months after surgery, and the mortality caused by different N stages tended to be the same after that. After 24 months, the biological characteristics of the tumor, such as Lauren classification, tumor location, and histological type, were more relevant to the prognosis of GC patients. Therefore, tumor site is not an independent prognostic factor for patients with early GC. As time goes by, the influence of the N stage on the tumor site also gradually decreases. This finding explains two problems. First, Verlato et al. [40] found that in the multivariate Cox analysis after 24 months of surgery, compared with N0, the HR value of N1 was less than 1, indicating that more lymph node metastasis had a protective effect on the prognosis of patients, consistent with this study. This indicates that the long-term prediction accuracy of N staging for patients has decreased over time, resulting in a phenomenon that does not meet theoretical expectations. Second, it further explains the advantages of hN staging over pN staging. hN staging incorporates the biological characteristics of the tumor. Therefore, in patients with longer survival time, although the prognostic advantage of N staging is reduced, the prognostic ability of the biological characteristics of the tumor is gradually increased. Hence, hN staging has improved the overall prognostic ability of patients compared with pN staging.

This staging system also has some limitations. Firstly, this is a retrospective study, and the sample recording process may be biased. In addition, this is a single-center study. More patients and testing centers could make this study more accurate and reliable. Secondly, the limitations of the hybrid staging system arise from its complexity. The system requires more preliminary preparation and staging calculation than the 8th AJCC staging system. This can increase the workload for surgeons and pathologists, who need to record the lymph node clearance. The hybrid staging system is also more complex than AJCC N staging, which utilizes only the number of positive lymph nodes. Physicians may require some study and practice to become proficient in its application.

## 5. Conclusions

Both the number and location of positive lymph nodes significantly impact the prognosis of patients with GC. The lymph nodes location-number hybrid staging system can improve the prognostic value of GC patients and enhance the correlation between the mixed staging and the tumor’s location and biological characteristics to improve the prediction efficiency. It can be used as a new installment instead of the 8th AJCC staging.

## Figures and Tables

**Figure 1 cancers-15-02659-f001:**
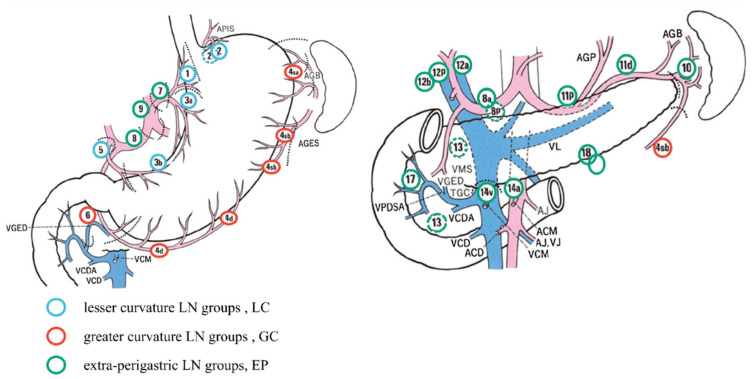
Extra-perigastric lymph node groups (EP), lesser curvature lymph node groups (LC), and greater curvature lymph node groups (GC) around the stomach. Figure source: Japanese Gastric Cancer Association (JGCA) [20].

**Figure 2 cancers-15-02659-f002:**
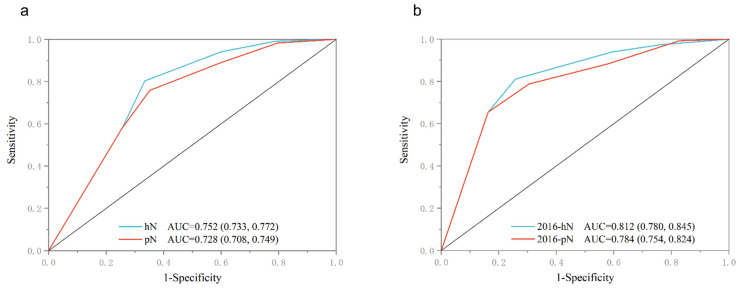
Comparison of AUC of hN and pN under N stage for training cohort and validation cohort, respectively. (**a**) AUC of training cohort, hN AUC = 0.752 (0.733, 0.772), pN AUC = 0.728 (0.708, 0.749). (**b**) AUC of validation cohort, 2016-hN AUC = 0.812 (0.780, 0.845), 2016-pN AUC = 0.784 (0.754, 0.824).

**Figure 3 cancers-15-02659-f003:**
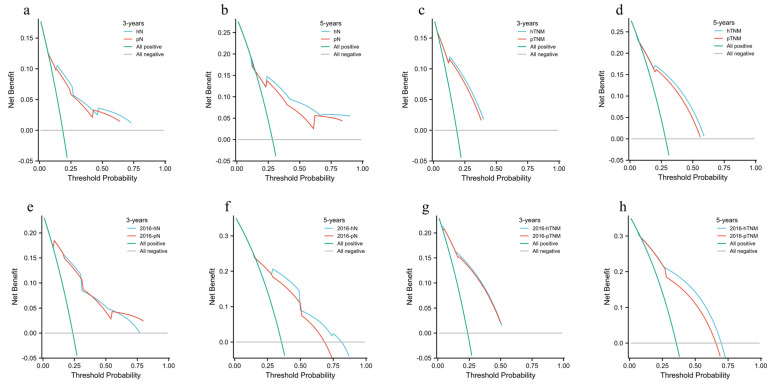
Comparison of DCA under N stage and TNM stage between the 3-year and 5-year hN staging and the 8th edition AJCC pathological staging in the training cohort. (**a**) Comparison of hN and pN DCA at 3 years in the training cohort. (**b**) Comparison of hN and pN DCA at 5 years in the training cohort. (**c**) Comparison of hTNM versus pTNM DCA at 3 years in the training cohort. (**d**) Comparison of hTNM versus pTNM DCA at 5 years in the training cohort. (**e**) Comparison of 2016-hN and 2016-pN DCA at 3 years in the validation cohort. (**f**) Comparison of 2016-hN and 2016-pN DCA at 5 years in the validation cohort. (**g**) Comparison of 2016-hTNM and 2016-pTNM DCA at 3 years in the validation cohort. (**h**) Comparison of 2016-hTNM and 2016-pTNM DCA at 5 years in the validation cohort.

**Figure 4 cancers-15-02659-f004:**
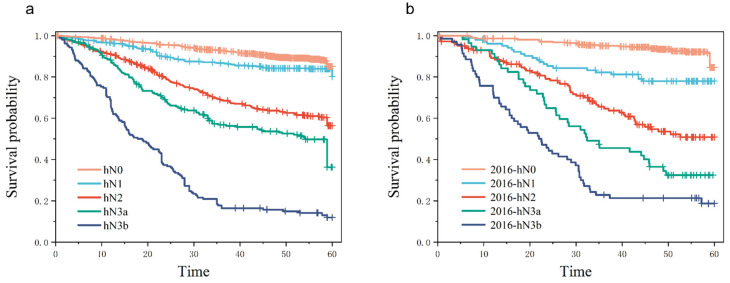
Kaplan–Meier curves of 5-year overall survival (OS) rates comparing subgroups according to the stage. (**a**) Survival curves of patients in each subgroup under the hN of the training cohort, with *p* < 0.001 between the subgroups. (**b**) Survival curves of patients in each subgroup under hN in the validation cohort, with *p* < 0.001 between the subgroups.

**Figure 5 cancers-15-02659-f005:**
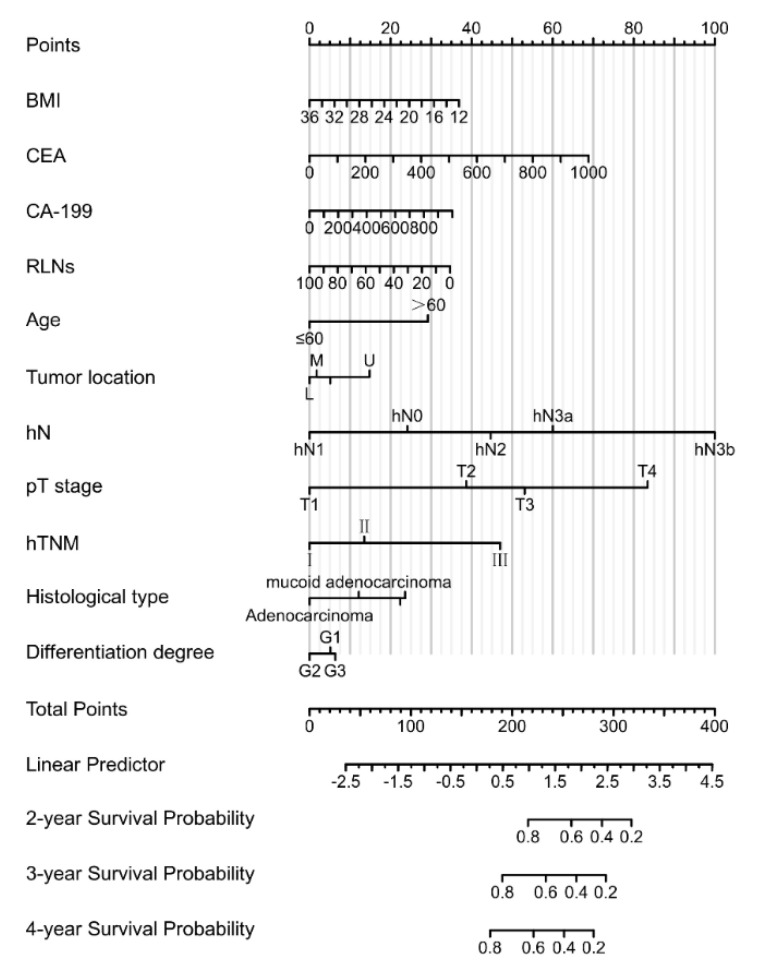
The prognostic Nomogram was established according to the independent prognostic factors, showing the relationship between the predictive variables in the prediction model.

**Table 1 cancers-15-02659-t001:** Patient baseline characteristics.

Characteristics	Total (%)	Training Cohort hN	Validation Cohort hN	*p*
*n*	3354	2598	756	
Age, *n* (%)				0.060
≤60	1899 (56.62%)	1494 (44.5%)	405 (12.1%)	
>60	1455 (43.38%)	1104 (32.9%)	351 (10.5%)	
Gender, *n* (%)				0.153
Male	2444 (72.87%)	1909 (56.9%)	535 (16%)	
Female	910 (27.13%)	689 (20.5%)	221 (6.6%)	
Tumor location, *n* (%)				0.383
L	2419 (72.13%)	1890 (56.4%)	529 (15.8%)	
M	530 (15.80%)	403 (12%)	127 (3.8%)	
U	354 (10.55%)	269 (8%)	85 (2.5%)	
LMU	51 (1.52%)	36 (1.1%)	15 (0.4%)	
hN, *n* (%)				<0.001
hN0	1671 (49.82%)	1291 (38.5%)	380 (11.3%)	
hN1	587 (17.50%)	483 (14.4%)	104 (3.1%)	
hN2	588 (17.53%)	443 (13.2%)	145 (4.3%)	
hN3a	280 (8.35%)	223 (6.6%)	57 (1.7%)	
hN3b	228 (6.80%)	158 (4.7%)	70 (2.1%)	
pT stage, *n* (%)				<0.001
T1	854 (25.47%)	656 (19.6%)	198 (5.9%)	
T2	527 (15.71%)	417 (12.4%)	110 (3.3%)	
T3	1218 (36.31%)	901 (26.9%)	317 (9.5%)	
T4	755 (22.51%)	624 (18.6%)	131 (3.9%)	
hTNM, *n* (%)				0.316
Ⅰ	1154 (34.41%)	885 (26.4%)	269 (8%)	
Ⅱ	1029 (30.68%)	814 (24.3%)	215 (6.4%)	
Ⅲ	1171 (34.91%)	899 (26.8%)	272 (8.1%)	
Histological type, *n* (%)				0.037
Adenocarcinoma	2679 (79.87%)	2090 (62.3%)	589 (17.6%)	
Low adhesion adenocarcinoma	460 (13.71%)	347 (10.3%)	113 (3.4%)	
mucoid adenocarcinoma	151 (4.50%)	120 (3.6%)	31 (0.9%)	
Signet-ring cell carcinoma	64 (1.92%)	41 (1.2%)	23 (0.7%)	
Differentiation degree, *n* (%)				0.031
G1	298 (8.88%)	221 (6.6%)	77 (2.3%)	
G2	1736 (51.76%)	1325 (39.5%)	411 (12.3%)	
G3	1320 (39.36%)	1052 (31.4%)	268 (8%)	
HER-2, *n* (%)				<0.001
0	1772 (52.83%)	1303 (38.8%)	469 (14%)	
1+	926 (27.61%)	758 (22.6%)	168 (5%)	
2+	434 (12.94%)	356 (10.6%)	78 (2.3%)	
3+	222 (6.62%)	181 (5.4%)	41 (1.2%)	
BMI, median (IQR)		22.59 (20.42, 24.8)	23.04 (20.76, 25.09)	0.029
CEA, median (IQR)		2.03 (1.2, 3.3)	1.89 (1.07, 3.04)	0.005
CA-199, median (IQR)		9.09 (5.6, 16.89)	10.02 (6, 16.89)	0.183
MLNs, median (IQR)		1 (0, 4)	0 (0, 5)	0.653
RLNs, median (IQR)		27 (20, 35)	27 (21, 34)	0.995
LNRS, median (IQR)		0.02 (0, 0.17)	0 (0, 0.19)	0.556
Tumor size, median (IQR)		40 (30, 60)	45 (30, 60)	0.042

**Table 2 cancers-15-02659-t002:** Baseline table of patients in the training cohort from 2011 to 2015.

Characteristics	Patients, *n* (%)	hN0, *n* (%)	hN1, *n* (%)	hN2, *n* (%)	hN3a, *n* (%)	hN3b, *n* (%)	*p*
*n*	2598	1291 (49.69%)	483 (18.59%)	443 (17.05%)	223 (8.58%)	158 (6.08%)	
Age, Mean ± SD	57.97 ± 10.05	57.61 ± 9.93	59.43 ± 10.02	57.79 ± 10.91	57.67 ± 9.23	57.42 ± 9.40	0.009
BMI, Mean ± SD	22.75 ± 3.22	23.12 ± 3.21	22.59 ± 3.15	22.37 ± 3.17	22.18 ± 3.19	22.18 ± 3.42	<0.001
CEA, Mean ± SD	6.64 ± 39.56	3.63 ± 14.93	6.45 ± 46.37	8.87 ± 42.58	7.45 ± 20.97	24.36 ± 106.42	<0.001
CA-199, Mean ± SD	30.78 ± 102.63	17.47 ± 58.88	26.04 ± 82.77	38.62 ± 113.76	55.37 ± 151.21	97.39 ± 225.42	<0.001
RLNs, Mean ± SD	28.21 ± 11.88	25.65 ± 10.83	27.84 ± 11.96	31.47 ± 12.47	29.73 ± 10.96	38.96 ± 11.04	<0.001
MLNs, Mean ± SD	3.62 ± 6.68	0	2.21 ± 2.08	6.87 ± 6.16	7.47 ± 2.77	22.97 ± 8.03	<0.001
LNRs, Mean ± SD	0.12 ± 0.19	0	0.09 ± 0.08	0.24 ± 0.20	0.28 ± 0.14	0.61 ± 0.18	<0.001
Age, *n* (%)							0.087
≤60	1494 (57.51%)	766 (59.33%)	254 (52.59%)	246 (55.53%)	134 (60.09%)	94 (59.49%)	
>60	1104 (42.49%)	525 (40.67%)	229 (47.41%)	197 (44.47%)	89 (39.91%)	64 (40.51%)	
Gender, *n* (%)							0.567
Male	1909 (73.48%)	943 (73.04%)	362 (74.95%)	316 (71.33%)	166 (74.44%)	122 (77.22%)	
Female	689 (26.52%)	348 (26.96%)	121 (25.05%)	127 (28.67%)	57 (25.56%)	36 (22.78%)	
Tumor location, *n* (%)							<0.001
L	1890 (72.75%)	974 (75.45%)	367 (75.98%)	311 (70.20%)	146 (65.47%)	92 (58.23%)	
M	403 (15.51%)	185 (14.33%)	60 (12.42%)	74 (16.70%)	50 (22.42%)	34 (21.52%)	
U	269 (10.35%)	123 (9.53%)	53 (10.97%)	45 (10.17%)	24 (10.76%)	24 (15.19%)	
LMU	36 (1.39%)	9 (0.70%)	3 (0.62%)	13 (2.93%)	3 (1.34%)	8 (5.07%)	
T stage, *n* (%)							<0.001
T1	656 (25.25%)	570 (44.15%)	61 (12.63%)	22 (4.97%)	3 (1.35%)	0 (0%)	
T2	417 (16.05%)	254 (19.67%)	85 (17.60%)	52 (11.74%)	21 (9.42%)	5 (3.16%)	
T3	901 (34.68%)	326 (25.25%)	197 (40.79%)	212 (47.86%)	104 (46.64%)	62 (39.24%)	
T4	624 (24.02%)	141 (10.92%)	140 (28.98%)	157 (35.43%)	95 (42.59%)	91 (57.60%)	
TNM stage, *n* (%)							<0.001
I	885 (34.06%)	824 (63.83%)	61 (12.63%)	0 (0%)	0 (0%)	0 (0%)	
II	814 (31.33%)	453 (35.09%)	282 (58.39%)	76 (17.16%)	3 (1.35%)	0 (0%)	
III	899 (34.61%)	14 (1.08%)	140 (29.98%)	367 (82.84%)	220 (98.65%)	158 (100%)	
Histological type, *n* (%)							<0.001
Adenocarcinoma	2090 (80.45%)	1051 (81.41%)	396 (81.99%)	358 (80.81%)	170 (76.23%)	115 (72.78%)	
Low adhesion adenocarcinoma	347 (13.36%)	187 (14.48%)	51 (10.56%)	50 (11.51%)	32 (14.35%)	27 (17.09%)	
Mucoid adenocarcinoma	120 (4.62%)	50 (3.87%)	21 (4.35%)	24 (5.42%)	16 (7.17%)	9 (5.70%)	
Signet-ring cell carcinoma	41 (1.57%)	3 (0.24%)	15 (3.10%)	11 (2.48%)	5 (2.24%)	7 (4.43%)	
Differentiation degree, *n* (%)							<0.001
G1	221 (8.51%)	144 (11.15%)	37 (7.66%)	23 (5.19%)	11 (4.93%)	6 (3.80%)	
G2	1325 (51.00%)	740 (57.32%)	223 (46.17%)	212 (47.86%)	101 (45.29%)	49 (31.01%)	
G3	1052 (40.49%)	407 (31.53%)	223 (46.17%)	208 (46.95%)	111 (49.78%)	103 (65.19%)	
HER-2, *n* (%)							0.028
0	1303 (50.15%)	686 (53.14%)	241 (49.90%)	197 (44.47%)	109 (48.88%)	70 (44.30%)	
1+	758 (29.18%)	364 (28.20%)	136 (28.16%)	136 (30.70%)	67 (30.04%)	55 (34.81%)	
2+	356 (13.70%)	172 (13.32%)	71 (14.70%)	67 (15.12%)	28 (12.56%)	18 (11.39%)	
3+	181 (6.97%)	69 (5.34%)	35 (7.24%)	43 (9.71%)	19 (8.52%)	15 (9.50%)	

**Table 3 cancers-15-02659-t003:** Baseline table of patients in the validation cohort in 2016.

Characteristics	Patients, *n* (%)	2016-hN0, *n* (%)	2016-hN1, *n* (%)	2016-hN2, *n* (%)	2016-hN3a, *n* (%)	2016-hN3b, *n* (%)	*p*
*n*	756	380 (50.26%)	104 (13.76%)	145 (19.18%)	57 (7.54%)	70 (9.26%)	
Age, Mean ± SD	59.10 ± 9.82	58.53 ± 9.99	60.34 ± 9.90	59.23 ± 9.81	60.46 ± 9.09	58.93 ± 9.40	0.398
BMI, Mean ± SD	23.01 ± 3.27	22.24 ± 3.06	23.36 ± 3.88	22.79 ± 3.27	23.18 ± 3.09	21.57 ± 3.17	0.001
CEA, Mean ± SD	4.80 ± 15.49	3.53 ± 12.36	3.66 ± 7.91	5.84 ± 11.78	5.54 ± 17.02	10.62 ± 33.42	0.022
CA-199, Mean ± SD	32.31 ± 112.96	18.65 ± 68.42	12.90 ± 54.05	45.01 ± 148.12	53.48 ± 147.16	75.23 ± 205.81	0.021
RLNs, Mean ± SD	28.10 ± 11.30	26.43 ± 11.15	27.40 ± 11.38	28.92 ± 10.32	28.75 ± 11.26	36.00 ± 10.73	<0.001
MLNs, Mean ± SD	3.63 ± 6.02	0	1.96 ± 1.54	6.05 ± 3.98	7.02 ± 1.91	18.01 ± 7.00	<0.001
LNRs, Mean ± SD	0.12 ± 0.19	0	0.09 ± 0.08	0.22 ± 0.15	0.27 ± 0.11	0.52 ± 0.17	<0.001
Age, *n* (%)							0.859
≤60	405 (53.57%)	208 (54.74%)	55 (52.88%)	76 (52.41%)	27 (47.37%)	39 (55.71%)	
>60	351 (46.43%)	172 (45.26%)	49 (47.12%)	69 (47.59%)	30 (52.63%)	31 (44.29%)	
Gender, *n* (%)							0.325
Male	535 (70.77%)	268 (70.53%)	74 (71.15%)	98 (67.59%)	47 (82.46%)	48 (68.57%)	
Female	221 (29.23%)	112 (29.47%)	30 (28.85%)	47 (32.41%)	10 (17.54%)	22 (31.43%)	
Tumor location, *n* (%)							<0.001
L	529 (69.97%)	280 (73.68%)	75 (72.12%)	103 (71.03%)	39 (68.42%)	32 (45.71%)	
M	127 (16.80%)	64 (16.84%)	16 (15.38%)	22 (15.17%)	8 (14.04%)	17 (24.29%)	
U	85 (11.24%)	35 (9.21%)	13 (12.50%)	15 (10.35%)	10 (17.54%)	12 (17.14%)	
LMU	15 (1.99%)	1 (0.27%)	0 (0%)	5 (3.45%)	0 (0%)	9 (12.86%)	
T stage, *n* (%)							<0.001
T1	198 (26.19%)	173 (45.53%)	16 (15.38%)	7 (4.83%)	1 (1.75%)	1 (1.43%)	
T2	110 (14.55%)	80 (21.05%)	11 (10.58%)	13 (8.97%)	5 (8.77%)	1 (1.43%)	
T3	317 (41.93%)	93 (24.47%)	60 (57.69%)	87 (60.00%)	35 (61.40%)	42 (60.00%)	
T4	131 (17.33%)	34 (8.95%)	17 (16.35%)	38 (26.20%)	16 (27.08%)	26 (37.14%)	
TNM stage, *n* (%)							<0.001
I	269 (35.58%)	253 (66.58%)	16 (15.38%)	0 (%)	0 (0%)	0 (0%)	
II	215 (28.44%)	123 (32.37%)	71 (68.27%)	20 (13.79%)	1 (1.75%)	0 (0%)	
III	272 (35.98%)	4 (1.05%)	17 (16.35%)	125 (86.21%)	56 (98.25%)	70 (100%)	
Histological type, *n* (%)							0.045
Adenocarcinoma	589 (77.91%)	310 (81.58%)	82 (78.85%)	111 (76.55%)	42 (73.68%)	44 (62.86%)	
Low adhesion adenocarcinoma	113 (14.95%)	48 (12.63%)	15 (14.42%)	21(14.48%)	9 (15.79%)	20 (28.57%)	
Mucoid adenocarcinoma	31 (4.10%)	14 (3.68%)	4 (3.85%)	8 (5.52%)	4 (7.02%)	1 (1.43%)	
Signet-ring cell carcinoma	23 (3.04%)	8 (2.11%)	3 (2.88%)	5 (3.45%)	2 (3.51%)	5 (7.14%)	
Differentiation degree, *n* (%)							<0.001
G1	77 (10.19%)	60 (15.79%)	9 (8.65%)	6 (4.14%)	1 (1.75%)	1 (1.43%)	
G2	411 (54.37%)	212 (55.79%)	44 (42.31%)	96 (66.21%)	26 (45.61%)	33 (47.14%)	
G3	268 (35.44%)	108 (28.42%)	51 (49.04%)	43 (29.65%)	30 (52.64%)	36 (51.43%)	
HER-2, *n* (%)							0.384
0	469 (62.04%)	242 (63.68%)	61 (58.65%)	86 (59.31%)	33 (57.89%)	47 (67.14%)	
1+	168 (22.22%)	87 (22.89%)	24 (23.08%)	33 (22.76%)	10 (17.54%)	14 (20.00%)	
2+	78 (10.32%)	36 (9.47%)	9 (8.65%)	16 (11.03%)	11 (19.30%)	6 (8.57%)	
3+	41 (5.42%)	15 (3.96%)	10 (9.62%)	10 (6.90%)	3 (5.27%)	3 (4.29%)	

**Table 4 cancers-15-02659-t004:** Comparison of C-index of hN and pN under different differentiation degrees, T stage, and TNM stage. (**a**) Comparison of C-index between hN and pN with different differentiation degrees. G1: highly differentiated. G2: moderately differentiated. G3: Poorly differentiated. (**b**) Comparison of C-index between hN and pN at different T stages. (**c**) Comparison of C-index between hN and pN in different TNM stages.

(a)				
**N**	**G1**	**G2**	**G3**	
hN	0.718	0.683	0.750	
pN	0.712	0.671	0.737	
**(b)**				
**N**	**T1**	**T2**	**T3**	**T4**
hN	0.550	0.703	0.742	0.674
pN	0.531	0.648	0.689	0.660
**(c)**				
**N**	**I**	**II**	**III**	
hN	0.520	0.623	0.621	
pN	0.520	0.611	0.606	

**Table 5 cancers-15-02659-t005:** Results of univariate and multivariate analyses using Cox proportional hazards model.

Characteristics	Total (N)	Univariate Analysis		Multivariate Analysis	
		Hazard Ratio (95%CI)	*p* Value	Hazard Ratio (95%CI)	*p* Value
BMI	3354	0.934 (0.915–0.954)	<0.001	0.971 (0.950–0.991)	0.006
CEA	3354	1.003 (1.002–1.004)	<0.001	1.002 (1.001–1.003)	0.002
CA-199	3354	1.002 (1.002–1.002)	<0.001	1.001 (1.000–1.001)	<0.001
MLNs	3354	1.075 (1.070–1.080)	<0.001	1.017 (0.994–1.041)	0.149
RLNs	3354	1.018 (1.013–1.023)	<0.001	0.991 (0.982–0.999)	0.030
LNRS	3354	13.647 (11.711–15.902)	<0.001	0.907 (0.388–2.122)	0.822
Tumor size	3354	1.019 (1.017–1.021)	<0.001	1.002 (0.999–1.005)	0.131
Age	3354				
≤60	1899	Reference			
>60	1455	1.714 (1.505–1.952)	<0.001	1.195 (1.002–1.425)	0.048
Gender	3354				
Male	2444	Reference			
Female	910	0.868 (0.747–1.009)	0.065	1.000 (0.857–1.166)	0.998
Tumor location	3354				
L	2419	Reference			
M	530	1.266 (1.063–1.507)	0.008	1.081 (0.901–1.296)	0.402
U	354	1.464 (1.205–1.780)	<0.001	1.320 (1.078–1.615)	0.007
LMU	51	3.554 (2.447–5.162)	<0.001	0.945 (0.617–1.447)	0.794
hN	3354				
hN0	1671	Reference			
hN1	587	1.359 (1.067–1.731)	0.013	0.612 (0.443–0.846)	0.003
hN2	588	4.516 (3.759–5.426)	<0.001	1.402 (0.945–2.078)	0.093
hN3a	280	6.890 (5.615–8.453)	<0.001	1.854 (1.210–2.843)	0.005
hN3b	228	16.234 (13.295–19.823)	<0.001	3.400 (2.051–5.637)	<0.001
pT stage	3354				
T1	854	Reference			
T2	527	2.822 (1.922–4.146)	<0.001	2.055 (1.339–3.153)	<0.001
T3	1218	7.811 (5.658–10.784)	<0.001	2.619 (1.509–4.546)	<0.001
T4	755	16.426 (11.921–22.633)	<0.001	4.643 (2.635–8.179)	<0.001
hTNM	3354				
I	1154	Reference			
II	1029	2.958 (2.255–3.881)	<0.001	1.285 (0.783–2.107)	0.321
III	1171	13.411 (10.526–17.085)	<0.001	2.483 (1.249–4.935)	0.009
Histological type	3354				
Adenocarcinoma	2679	Reference			
Low adhesion adenocarcinoma	460	1.367 (1.144–1.633)	<0.001	1.547 (1.283–1.866)	<0.001
mucoid adenocarcinoma	151	1.598 (1.220–2.094)	<0.001	1.176 (0.892–1.548)	0.250
Signet-ring cell carcinoma	64	2.610 (1.840–3.701)	<0.001	1.565 (1.094–2.239)	0.014
Differentiation	3354				
G1	298	Reference			
G2	1736	1.546 (1.151–2.078)	0.004	0.890 (0.659–1.202)	0.448
G3	1320	2.223 (1.655–2.987)	<0.001	1.009 (0.744–1.369)	0.953
HER-2	3354				
0	1772	Reference			
1+	926	1.002 (0.859–1.168)	0.981		
2+	434	1.042 (0.854–1.271)	0.688		
3+	222	1.274 (0.996–1.630)	0.054		

## Data Availability

Due to patient privacy, no Data Availability Statement was provided for this article.

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
