# Peer review of "Efficacy of Lymph Node Location-Number Hybrid Staging System on the Prognosis of Gastric Cancer Patients"

_cancers, 2023, doi:10.3390/cancers15092659_

Round 1
Reviewer 1 Report
Dear Author, really interesting manuscript about staging system of lymph node location and number in gastric cancer patients.
Good quality of language and statistical analysis.
Good job
Author Response
Dear Reviewer,
Thank you for taking the time to review my manuscript. I am grateful for your positive feedback and encouragement. I have had my manuscript professionally edited to ensure that there are no errors in English expression or grammar. I have attached the certificate of English editing for your reference (filename: Certificate of English Editing.pdf).
Thank you again for your thoughtful review and suggestions. I look forward to the opportunity to publish this work in your esteemed journal.
Wishing you good health.
Best regards,
Junpeng Wu

Reviewer 2 Report
Thank you for sending me the research article paper “Lymph node location-number hybrid staging system on the prognosis of gastric cancer patients” for review in the Cancers. In the article of Wu et al., the author proposed a new method for the prognosis of lymph node associated gastric cancer. There are important points that should be discussed and improved.
1. What is the standard of hybrid staging systems for the prognosis of gastric cancer? It is a new method of staging. Hence, the author should explain the standard and limitations of hybrid systems in detail.
2. Authors should provide the inclusion and exclusion criteria of samples.
Author Response
Dear Reviewer,
Thank you for reviewing my manuscript and providing valuable feedback. I have addressed your concerns in the revised version of the manuscript, which is attached as "Response to Reviewer 2 Comments.docx".
Additionally, I want to inform you that we have had the manuscript professionally edited to ensure there are no English language or grammar errors. The certificate of English editing is attached in the same attachment along with the revised manuscript.
Thank you again for your time and effort in reviewing my manuscript. I hope that the changes I have made will improve the scientific quality of the manuscript, and I look forward to the possibility of its publication.
Best regards,
Junpeng Wu

Reviewer 3 Report
The manuscript "Lymph node location-number hybrid staging system on the prognosis of gastric cancer patients" presents an extensive retrospective study of Asian gastric cancer patients evaluated according to a proposed hybrid TNM staging, based on the location of the metastatic lymph nodes.
The study represents a valuable contribution to the literature, but needs improvement and supplementary correlations in future publications. Although the authors considered a flaw the retrospective design of the study, I would have considered it beneficial and an opportunity for determining the OS of the patients, the PFS, DFS, more accurate and relevant in terms of prognosis.
No reference to the treatment was made, except to the surgical approach.
I suggest making reference to the changes of the treatment course that this hybrid staging could bring in order to improve the patient survival.
I recommend changing the approach of the conclusion, making it more open to the fact that an hN could determine a better prognosis in correlation with a pN.
Best of luck!
Author Response
Dear Reviewer,
Thank you for taking the time to review my manuscript. I am grateful for your positive feedback and encouragement. I have had my manuscript professionally edited to ensure that there are no errors in English expression or grammar. I have attached the certificate of English editing for your reference (filename: Certificate of English Editing.pdf). Thank you for your suggestions on the manuscript. In future research, we will include OS, PFS, DFS, and other data to enhance the accuracy of the paper.
Point 1: The Reviewer suggest that data such as PFS and DFS be added to the results
Response 1: Thank you for your understanding. I apologize for the limitation in obtaining data on PFS and DFS for the 4712 patients included in our study. We acknowledge the importance of these parameters and their relevance to our research, but unfortunately it is difficult for us to refollow the 4712 patients included to determine PFS or DFS. However, this is still an important research value and we will focus on this aspect in future studies
Point 2: No reference to the treatment was made, except to the surgical approach.
Response 2: All patients did not receive preoperative neoadjuvant chemotherapy, chest and abdomen enhanced CT/MR Were routinely examined before surgery, and postoperative analgesia and anti-inflammatory therapy were routinely performed when no distant metastasis was diagnosed by imaging, and drainage was generally removed on the fifth day after surgery.
Point 3: I recommend changing the approach of the conclusion, making it more open to the fact that an hN could determine a better prognosis in correlation with a pN.
Response 3: Both the number and location of positive lymph nodes significantly impact the prognosis of patients with GC. The Lymph nodes location-number hybrid staging system can improve the prognostic value of GC patients and enhance the correlation between the mixed staging and the tumor's location and biological characteristics to improve the prediction efficiency. hN staging showed better results in ROC, DCA and C-index than AJCC pN staging. The results showed that hN had a significant advantage over pN stage in predicting the prognosis of patients.

Round 2
Reviewer 2 Report
Accepted
Reviewer 3 Report
Thank you for the modifications made according to the suggestion.